

# Divergent effect of fluoxetine on the response to physical or chemical stressors in zebrafish

Murilo S. Abreu[1],*, Ana Cristina V.V. Giacomini[1,2],*, Gessi Koakoski[1],*, Angelo L.S. Piato[3],* and Leonardo J.G. Barcellos[1,4],*

[1] Programa de Pós-Graduação em Farmacologia, Universidade Federal de Santa Maria, Santa Maria, Rio Grande do Sul, Brazil
[2] Universidade de Passo Fundo, Passo Fundo, Rio Grande do Sul, Brazil
[3] Programa de Pós-Graduação em Farmacologia e Terapêutica, Universidade Federal do Rio Grande do Sul, Porto Alegre, Rio Grande do Sul, Brazil
[4] Programa de Pós-Graduação em Bioexperimentação, Universidade de Passo Fundo, Passo Fundo, Rio Grande do Sul, Brazil
* These authors contributed equally to this work.

## ABSTRACT

Fluoxetine is a selective serotonin reuptake inhibitor that increases serotonin concentration in the central nervous system and modulates various systems, including the control of sympathetic outflow and the hypothalamus–pituitary–adrenal. However, it is not yet established whether fluoxetine can modulate the responses to stressors stimulants (physical or chemical) that trigger cortisol response in zebrafish. We demonstrate that fluoxetine blunts the response to physical stress, but not to chemical stress.

## INTRODUCTION

Fluoxetine (FLU), a selective serotonin reuptake inhibitor (SSRI), increases serotonin concentration in the central nervous system (*Wong, Bymaster & Engleman, 1995*). Serotonin is one of the major neurotransmitters in the central nervous system and modulates various systems, including the control of sympathetic outflow and the hypothalamus–pituitary–adrenal axis (HPA), via serotonergic fibers that innervate structures such as the hippocampus, prefrontal cortex, amygdala, and hypothalamus (*Lowry, 2002*). SSRIs and cognitive–behavioral therapy are both effective treatments for generalized anxiety disorder, and are known to reduce the peak of cortisol in older adults (*Rosnick et al., 2016*). FLU has been shown to blunt the cortisol response (*Abreu et al., 2014*) and, as a consequence, prevent stress-related osmoregulation changes in zebrafish (*Abreu et al., 2015*). In addition, fluoxetine reverses the anxiogenic effects of acute (*Giacomini et al., 2016*) and chronic (*Marcon et al., 2016*) stress in this species.

Stress depends on a stressor stimulus to occur, and in mammals it triggers a stimulatory process in the hippocampus and amygdala (*LeDoux, 2000, 2007*). In the hypothalamus, stress stimulates the release of corticotropin-releasing factor, which is the key

Corresponding author
Leonardo J.G. Barcellos, lbarcellos@upf.br

neurotransmitter regulating the release of adrenocorticotropic hormone from the pituitary, which in turn induces the release of glucocorticoids (cortisol) from the adrenal. In teleost fish like in zebrafish, the hypothalamic–pituitary–interrenal axis is the HPA axis homolog (Wendelaar Bonga, 1997).

Stress stimuli can be varied (e.g., social, physical, chemical), such as exposure to neighborhood-level violence, which can influence physiological and cellular markers of stress, even in children (Theall et al., 2017). In addition, physical stimuli elicit robust stress responses in fish (Perry, Reid & Salama, 1996). Physical stressors such as chasing have been used as standardized stressors (Abreu et al., 2014; Giacomini et al., 2015, 2016), and spatial restriction is used as a stress model for behavioral assessment in zebrafish (Piato et al., 2011; Ghisleni et al., 2012). Stressor stimulus can also be chemical, such as alarm substances, originally described in the minnow (Phoxinus phoxinus) (Frisch, 1941), which are produced and stored in epidermal "club" cells (Barbosa et al., 2012) and are released into the water after skin injuries as those provoked by predator attack (Chivers & Smith, 1998; Korpi & Wisenden, 2001). Alarm substance is known to induce fear responses in a range of fish species (Pfeiffer, 1977). Moreover, blood (Barreto et al., 2013) and diamines (putrescine and cadaverine) (Hussain et al., 2013) have also been documented as potential chemical stressors. However, it is not yet established whether FLU can modulate the responses to different modalities of stressor stimuli (physical or chemical) that trigger cortisol response in zebrafish.

## MATERIALS AND METHODS

### Experimental animals

A stock population of 200 mixed-sex (50/50) 180-day-old wild-type zebrafish (Danio rerio), weighing 0.45 ± 0.05 g, short-fin (SF) strain, was maintained in two tanks equipped with biological filters, under constant aeration, and with a natural photoperiod (approximately 14 h light:10 h dark). Water temperature was maintained at 26 ± 1 °C; pH at 7.0 ± 0.2; dissolved oxygen at 6.1 ± 0.2 mg/L; total ammonia at <0.01 mg/L; total hardness at 6 mg/L; and alkalinity at 22 mg/L $CaCO_3$. This study was approved by the Ethics Commission for Animal Use (CEUA) of Universidade de Passo Fundo, Passo Fundo, Rio Grande do Sul, Brazil (Protocol #29/2014-CEUA) and met the guidelines of Conselho Nacional de Controle de Experimentação Animal (CONCEA).

### Experimental protocol

Our aim was to verify whether FLU modulates cortisol changes induced by physical and chemical stressors in zebrafish. After a 15-day period for acclimation to laboratory conditions, fish were randomly distributed into two groups, i.e., untreated fish (control group) and fish exposed to FLU. The latter group was exposed to FLU (Daforin®, EMS, Brazil, São Bernardo do Campo) at a concentration of 50 μg/L for 15 min. before the stressor stimuli (Fig. 1); this concentration and duration of exposure were previously shown to elicit behavioral responses (Giacomini et al., 2016) and decrease cortisol response in acute chasing stress (Abreu et al., 2014).

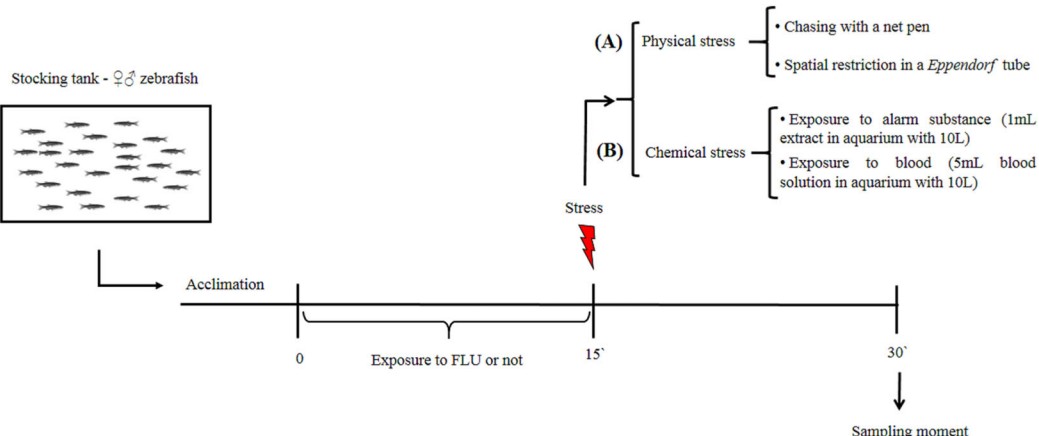

**Figure 1 Schematic representation of the experimental design.**

### Physical stimuli on stress response

To evaluate the physical stress response, we then subdivided control and treated fish into groups of 10 animals (duplicate) that were submitted or not to the following types of physical stress: chasing with a net (duration 2 min, and waiting to complete 15 min to sampling); spatial restriction in a microtube (duration 15 min) (Fig. 1A). After the 15 min of exposure to each stressor, fish were captured, euthanized by decapitation with medulla sectioning and immediately frozen in liquid nitrogen for storage at −80 °C until cortisol extraction (Fig. 1). This time interval was based on previous studies showing that cortisol levels peak 15 min following presentation of a stressor stimulus (*Abreu et al., 2014*; *Idalencio et al., 2015*; *Ramsay et al., 2009*).

### Chemical stimuli on stress response

To evaluate the chemical stress response, we then subdivided control and treated fish into groups of 10 animals (duplicate) that were submitted or not to the following types of chemical stress: exposure to conspecific blood (duration 15 min); and exposure to alarm substance of conspecifics (duration 15 min). Exposure to blood (5 mL, extracted from zebrafish and jundia (*Rhamdia quelen*)—the use of jundia blood was due to the low yield of zebrafish blood extraction) was in a 10 L aquarium (*Barreto et al., 2013*); and exposure to alarm substance of conspecifics (*Speedie & Gerlai, 2008*) (1 mL, zebrafish) was in a 10 L aquarium (*Barreto et al., 2010*). After 15 min of exposure to each stressor, fish were captured, euthanized, and stored as described above (Fig. 1B). For collection of fish blood (zebrafish and jundia), fish were anesthetized by eugenol (400 mg/L), the anesthesia occurred in less than 1 min and determined by total loss of opercular movement followed by cardiac arrest; then the caudal peduncle was sectioned for the collection of blood. For extraction of alarm substance, fish were quickly killed by medulla sectioning, then shallow cuts were made on each side of fish and the cuts were washed with distilled water; at the end of the process a total of 100 mL of alarm substance in solution were collected (*Speedie & Gerlai, 2008*).

## Cortisol analysis

Whole-body cortisol levels were determined using the method described by *Sink, Kumaran & Lochmann (2007)*. Fish were weighed, minced, and homogenized with phosphate buffered saline (pH 7.3). Samples were transferred into tubes with ether, vortexed, centrifuged, and then immediately frozen in liquid nitrogen (three times this last process). The unfrozen portion (ethyl ether containing cortisol) was decanted and transferred to a new tube and completely evaporated, yielding a lipid extract containing the cortisol. The samples were then placed on the plate of enzyme-linked immunosorbent assay kit. The accuracy was tested by calculating the recoveries from samples spiked with known amounts of cortisol (50, 25, and 12.5 ng/mL), the mean detection of spiked samples was 94.3%. All cortisol values were adjusted for recovery with the following equation: cortisol value = measured value × 1.0604. Whole-body cortisol levels were measured in duplicate for each extraction using the commercially available enzyme-linked immunosorbent assay kit (EIAgen CORTISOL test, BioChem Immunosystems, Rome, Italy). Reading was carried out in microplate reader equipment (ASYS UVM 340, ASYS, Chorley, UK).

## Statistical analysis

After testing the homogeneity of variance and normality of data (Hartley and Kolmogorov–Smirnov tests, respectively), we compared the whole-body cortisol levels using two-way analysis of variance (ANOVA) followed by Dunnett's post hoc test. Differences were considered statistically significant at $p < 0.05$. The data are expressed as mean + SEM.

## RESULTS

### Physical stimuli on stress response

Fish exposed to physical stressors (spatial restriction or chasing) displayed an increase in cortisol levels, and FLU blunted the increase in cortisol levels in fish subjected to physical stressors (Fig. 2). Two-way ANOVA revealed significant interaction between the factors ($F_{2, 45} = 6.080$, $p = 0.0046$), main effects of drug ($F_{1, 45} = 13.89$, $p = 0.0005$) and stress ($F_{2, 45} = 12.93$, $p < 0.0001$).

### Chemical stimuli on stress response

Fish exposed to chemical stressors (alarm substance or blood) displayed an increase in cortisol levels, but FLU did not blunt the increase in cortisol levels in fish subjected to chemical stressors (Fig. 3). Two-way ANOVA revealed a significant main effect of stress ($F_{2, 48} = 5.623$, $p = 0.0064$), but not interaction effect between the factors ($F_{2, 48} = 0.7045$, $p = 0.4994$) or a main effect of drug ($F_{1, 48} = 0.01718$, $p = 0.8963$).

## DISCUSSION

Here we show that fluoxetine blunts the response to physical, but not chemical, stress. Whether physical (*Ramsay et al., 2009*) or chemical (*Teles et al., 2017*) stress increases cortisol levels in zebrafish.

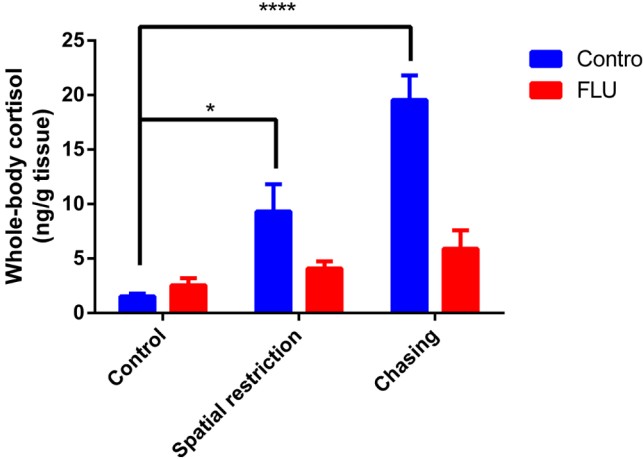

**Figure 2 Effects of physical acute stressors (spatial restriction or chasing) on cortisol levels in whole-body zebrafish.** Data were expressed as mean + SEM. Two-way ANOVA followed by Dunnett's post hoc test. FLU (fluoxetine). $^*p < 0.05$ and $^{****}p < 0.0001$.

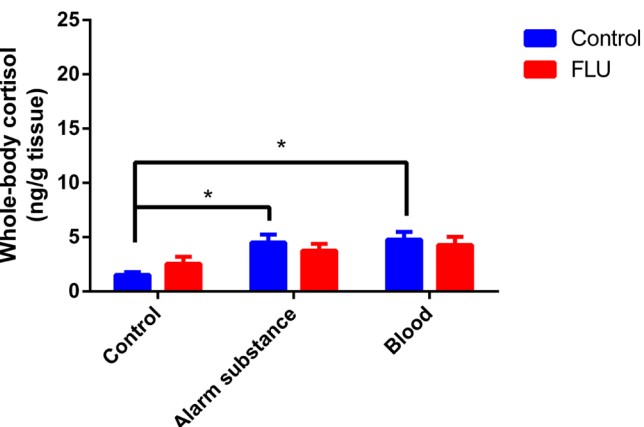

**Figure 3 Effects of chemical acute stressors (alarm substance or blood) on cortisol levels in whole-body zebrafish.** Data were expressed as mean + SEM. Two-way ANOVA followed by Dunnett's post hoc test. FLU (fluoxetine). $^*p < 0.05$.

The greater magnitude of response to a physical stressor could be related to its high impact can cause a clear aversive response in fish (*Abreu et al., 2016*). Besides, confinement stress also resulted in elevated cortisol for being "high-impact stress" (*Silva et al., 2015*), perhaps physical stressors act in dorsolateral and dorsomedial regions of the pallium that have been characterized as functional homologues to the mammalian amygdala and hippocampus (*Goodson & Kingsbury, 2013*; *O'Connell & Hofmann, 2011*; *Vargas, López & Portavella, 2009*), with consequent action under the hypothalamus. On the other hand, chemical stress does not trigger a response of such magnitude (*Silva et al., 2015*). Our hypothesis is that the chemical stressor stimulus depends on more than one sensory pathway (e.g., smell, tactile) for the perception of the stimulus, which would result in a suppression of the stimulation force of the hypothalamic system, with consequent pituitary and later adrenergic stimulation.

We demonstrated that fluoxetine prevents the increase of cortisol in fish in response to physical stressor stimulus. Previously, we showed that fluoxetine blocked cortisol response to acute chasing stress in a dose-dependent manner (*Abreu et al., 2014*) as well as in fish subjected to different forms of housing (*Giacomini et al., 2016*). FLU also blocked the stress response following chronic exposure in zebrafish (*Egan et al., 2009*), besides stress increases serotoninergic activity in the telencephalon in fish (e.g., *Øverli et al., 2004*; *Winberg, Nilsson & Olsen, 1992*). In fact, the levels of serotonin in the brain regions considered homologous to the mammalian hippocampus and amygdala are altered in fish subjected to spatial restriction (*Silva et al., 2015*). This effect reinforces the participation of these regions in response to physical stress, as well as the involvement of serotonin in these pathways.

Still, we have shown that fluoxetine did not block the increase of cortisol in fish in response to chemical stressor stimulus. Alarm substance induced stress responses in *Nile tilapia* (*Oreochromis niloticus*), increasing ventilation rate and cortisol level (*Sanches et al., 2015*) as well as increasing erratic movements in zebrafish (*Speedie & Gerlai, 2008*). The exposure to blood has also been shown to induce antipredator behavior in the fish species *N. tilapia* (*Barreto et al., 2013*). The exposure to alarm substance also increased anxiety-like behavior in the light/dark test in zebrafish and decreased nocifensive behavior, however pretreatment with fluoxetine blocked the anxiogenic effects of alarm substance on the light/dark test and also increased extracellular brain 5-HT (*Maximino et al., 2014*), the same behavioral relationship between alarm substance and serotoninergic system was not observed in the relationship between neuroendocrine and serotoninergic system. Serotonin receptors ($5\text{-HT}_{1A}$ and $5\text{-HT}_4$) expressed in steroidogenic cells in the interrenal glands mediate the effects of serotonin on cortisol response (*Herculano & Maximino, 2014*), and this direct mechanism may underlie the effects of fluoxetine observed in physical stress response, namely the inhibition of cortisol release.

### Funding

This study was funded by the Universidade de Passo Fundo and CNPq (grant number 470260/2013-0). LJGB holds a CNPq research fellowship (301992/2014-2). The funders had no role in study design, data collection and analysis, decision to publish, or preparation of the manuscript.

### Grant Disclosures

The following grant information was disclosed by the authors:
Universidade de Passo Fundo and CNPq: 470260/2013-0.
CNPq research fellowship: 301992/2014-2.

### Competing Interests

Angelo L.S. Piato is an Academic Editor for PeerJ.

## Author Contributions

- Murilo S. Abreu conceived and designed the experiments, performed the experiments, analyzed the data, wrote the paper, prepared figures and/or tables.
- Ana C.V.V. Giacomini conceived and designed the experiments, performed the experiments, analyzed the data and wrote the paper.
- Gessi Koakoski contributed reagents/materials/analysis tools.
- Angelo L.S. Piato wrote the paper, reviewed drafts of the paper.
- Leonardo J.G. Barcellos conceived and designed the experiments, analyzed the data, wrote the paper, reviewed drafts of the paper.

## Animal Ethics

The following information was supplied relating to ethical approvals (i.e., approving body and any reference numbers):

This study was approved by the Ethics Commission for Animal Use (CEUA) of Universidade de Passo Fundo, Passo Fundo, Rio Grande do Sul, Brazil (Protocol #29/2014-CEUA) and met the guidelines of Conselho Nacional de Controle de Experimentação Animal (CONCEA).

## Data Availability

The raw data has been supplied as Supplemental Dataset Files.

## Supplemental Information

Supplemental information for this article can be found online at http://dx.doi.org/10.7717/peerj.3330#supplemental-information.

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
