# Peer review of "Divergent effect of fluoxetine on the response to physical or chemical stressors in zebrafish"

_PeerJ, doi:10.7717/peerj.3330_

## Round 0.1 · original submission · Major Revisions

This paper reports on the effects of fluoxetine on cortisol levels in zebrafish exposed to a variety of stressors. The major flaw in the manuscript is the failure of the authors to outline what is new about the results of the experiments. They state in the Abstract (and repeat in the Introduction) "it is not yet established whether (physical or chemical) stimulants trigger cortisol response in zebrafish, and whether fluoxetine can modulate the responses to these stimuli" yet this is contradicted by their own previous published work (referred to in the paper, including the Methods Section where they explain why they chose doses of drug and timepoints of cortisol measurement). Many other labs have also shown cortisol responses to stress in zebrafish. No clear case is made for what is novel in this study, or why repeating previous work was necessary.

The reviewers raise a number of issues about the design and analysis of the experiments and the language of the manuscript, and these would also need to be addressed fully in any revised submission. I have some additional comments:

- The use of liquid nitrogen alone as a method of euthanasia needs some justification – published guidelines refer to immobilization with MS-222 followed by immersion in liquid nitrogen as being appropriate, but what is the evidence that direct immersion in liquid nitrogen is (relatively) painless ?

- How was the zebrafish and catfish blood collected ? How many zebrafish were used to collect 5 mls of blood – presumably quite a few ?

- What are “alarm substances” and how were they collected ?

·

Basic reporting

The writing style is relatively clear and concise.
There are just a few places where the use of English is slightly incorrect, which may benefit from final proof-reading by a Native English speaking colleague. Minor changes will be able to fix these small issues and improve the manuscripts readability.
Some examples of places to improve include:
-'persecution' is not a particularly scientific word to use for the chasing of an animal with a net. Is this not 'chasing' as described within Abreu et al 2014.
-Line 28: 'Therefore' doesn't belong at the beginning of this sentence and '(physical or chemical)' belongs after the word 'stimulants' not in front of it.

Summary of the literature appears to be good.
Perhaps put the statement of ethical clearance at the beginning of the 'Experimental animals' section, rather than at the end.

Experimental design

Overall the experimental design is fine.
I feel that the submission does not fill a knowledge gap in the literature. It appears to be a close replication of work previously performed by the same group previously (Abreu et al. 2014). In Abreu et al (2014) the authors also found that stress (induced by 2 mins of net chasing) caused zebrafish to have elevated cortisol levels and that fluoxetine treatment in the water (15 mins prior) ‘blunted’ that increase.

If this submission is more than a repeat of the researchers previous findings the research question needs to be clarified within this submission so that it is clear that these findings will make a meaningful contribution. For example, perhaps more of a comparison between the effects of physical and chemical induced stress and the effects of fluoxetine therein should be discussed.

Ethical standards: I hope that the authors have ethical approval for the spatial restriction protocol because confining a 180 day old zebrafish within an eppendorf tube (as indicated in Figure 1) would need justification in my opinion. What size (volume) eppendorf tube is it?

Validity of the findings

I find the statistical findings hard to follow because not all significant effects are indicated on the graphs shown. Also, on line 164 it says there is a significant effect, but the p-value listed is p=0.4994 and in line 165 it says there was no significant effect of stress but the p-value listed is =0.0064.
Within the abstract it is concluded that 'fluoxetine blunts the response to stress, regardless of its nature, chemical or physical' however the statistical finding on lines 164-5 says that there was no statistical effect of drug (fluoxetine) in the chemical stress studies.

Reviewer 2 ·

Basic reporting

The English language needs significant improvement. The numerous grammatical errors, awkward phrasing and typos made comprehension very difficult. The lack of attention to details during the preparation for this did not give a professional impression of the authors. In its current state, the manuscript is very much a draft and should not have been submitted for review.

Justification/context for the study is unconvincing. The numerous refereniced studies contradict the author's claim that it has not yet been established that physical or chemical stimulant can trigger cortisol response in zebrafish. A quick search of the current literature, including the author's previous works, would suggest otherwise. If the objective of the study was to compare the effects of fluoxetine on cortisol response between physical and chemical stimuli, the concept was not clearly communicated in the introduction. This is a descriptive study showing that acute exposure to fluoxetine blunts cortisol responses induced by 4 different stimuli with very little mechanistic explanation provided.

Experimental design

Average wt of the animal should be reported.

It was unclear whether exposure to fluoxetine was done at an individual basis, prior to subjection to stress stimuli, or as a group.

Chemical analysis to confirm the nominal concentrations of fluoxetine should be provided.

Sample processing for whole body cortisol measurements should be mentioned even if briefly.

The model and manufacturer of the ELIZA reader should be stated.

It is unclear how many individuals were analyzed per treatment group for whole body cortisol. From the figures, it seems like 12 were used for spatial restriction for the control group and and 9 were measured for the control group for blood. Are the bars SEM or SD?

Validity of the findings

There were no differences between whole-body cortisol levels between control and flu after persecution? If so then why was the effect of the drug significant (p=0.0005)?

The text does not reflect what is shown in the figure (line 164-165). Stress does have an effect.

The discussion is extremely difficult to read and this is due to both the use of the English language as well as the organization and structure of the writing. It is unclear to the reviewer what the significance of the findings was at the end.

Line 174-177 directly contracts the novelty of the research as stated by the authors in both the abstract and introduction.

Line 185-188: please clarify what "only the similar tendency between physical stressor and fish exposed to chemical cues" means?

The hypothesis stated in the discussion is not being supported.

---

## Round 0.2 · Major Revisions

Both a reviewer and I continue to have concerns about the methods of euthanasia used in this Ms. In particular, the euthanasia of unanaesthetized adult zebrafish requires justification - can the authors please supply guidelines indicating that decapitation (if that is what happened) is acceptable - under the AVMA Guidelines it probably would not be. The eugenol anaesthesia also needs to be more fully described - for how long, and how was its anaesthetic effect determined before the procedure.

·

Basic reporting

The use of English language is improved, however is still use of some unusual phrases. For example, the expression 'stressors stimulants' is used frequently, 'stressor stimuli' might be more correct.
Results are relevant to the hypothesis.

Figure 2. why is the significant effect of FLU treatment not indicated by a symbol?

Experimental design

The research question is now more clearly defined and shows an apparent knowledge gap (comparison of effect of FLU on physical and chemical stress).

Ethics standards: I am not convinced that all of the work meets international ethical standards, for example: use of 'medulla section' (decapitation?) for euthanasia without prior anaesthesia or pithing follow-up (both recommended by American Veterinary Medical Association). Further details on the duration of application of eugenol to ensure euthanasia prior to blood collection need to be included. Ethical standards need to be considered and further details need to be provided if they have been omitted, otherwise justification needs to be provided.

Validity of the findings

This is fine.

---

## Round 0.3 · accepted · Accept

Thanks for responding to my concerns. For what it is worth, I don't agree that you have followed the AVMA Guidelines, as they explicitly specify pithing of zebrafish after decapitation. There also comments about the use of eugenol in the Guidelines that you should have a careful look at. I appreciate that you have local ethics approval, and that ethical norms are not uniform, however, I would urge you to carefully consider your experimental practice in the future.